# Voronoi Diagrams for Senior-Friendly Cities

**DOI:** 10.3390/ijerph19127447

**Published:** 2022-06-17

**Authors:** Marta Figurska, Agnieszka Dawidowicz, Elżbieta Zysk

**Affiliations:** 1Department of Socio-Economic Geography, Institute of Spatial Management and Geography, Faculty of Geoengineering, University of Warmia and Mazury in Olsztyn, 10-719 Olsztyn, Poland; 2Department of Land Management and Geographic Information Systems, Institute of Spatial Management and Geography, Faculty of Geoengineering, University of Warmia and Mazury in Olsztyn, 10-719 Olsztyn, Poland; 3Department of Spatial Analysis and Real Estate Market, Institute of Spatial Management and Geography, Faculty of Geoengineering, University of Warmia and Mazury in Olsztyn, 10-719 Olsztyn, Poland; elzbieta.zysk@uwm.edu.pl

**Keywords:** Voronoi diagrams, senior-friendly city, active aging, older people, GIS, Poland

## Abstract

Motives: Active aging places (AAP) should be identified during the COVID-19 pandemic to ensure the sanitary safety of seniors, prevent older adults from feeling excluded, and eliminate health threats that discourage seniors from being active. Aim: The aim of this study was to apply a new analytical approach with the use of Voronoi diagrams in GIS tools to spatially identify the AAP in the context of apparent social dynamics. Methods: An empirical study was conducted with the use of qualitative (literature review, questionnaire survey, AAP classification, visualization of AAP location with GIS tools) and quantitative methods (AAP ranking based on a statistical analysis of survey responses). Results: Voronoi diagrams were used to determine the accessibility of selected objects in the city of Olsztyn and identify spaces that belong to the social logic of space.

## 1. Introduction

Population aging, which is a characteristic of highly-developed countries, has become a clear global trend over the past two decades. This phenomenon is caused by various demographic, economic, civilizational, and cultural factors. The most important of these are low and declining birth rate, high and increasing life expectancy, health promotion and preventive medicine, as well as effective social security, pension, and retirement systems [1]. According to World Bank data [2], the world’s population more than doubled between 1960 and 2020, from 3.032 to 7.753 billion. At the same time, the proportion of seniors in the world’s population has increased from 5% (1960) to more than 9.3% (2020) [3]. This indicator nearly doubled, which clearly indicates that the global society is aging. This phenomenon should be monitored to adapt the existing urban planning solutions to the needs of the growing senior population and create friendly spaces that promote active aging [4]. The “Age-Friendly Cities” document [5] emphasizes the need to establish cities that account for the needs of the oldest citizens in six key aspects: housing; community support and health services; transportation; communication and information; outdoor spaces and buildings; social participation, and; civic participation, respect, and social inclusion. These goals were also accentuated by the Agenda for Human Rights in the City [6], which states that every resident should feel welcome and comfortable in a city. These documents coincide with the Sustainable Development Goals [7] for panel 11 “Sustainable cities and communities (make cities and human settlements inclusive, safe, resilient and sustainable)”, particularly in the context of examining the quality of life in cities, as well as standard ISO 37120:2018 “Sustainable cities and communities—Indicators for city services and quality of life” [8]. All of them aim at shaping older people’s living spaces in a way that would promote active aging [9]. According to the definition proposed by the World Health Organization [9], active aging is the process of optimizing opportunities for health, participation, and security in order to enhance the quality of life as people age. Sustainable management of cities should be based on a spatial policy that accounts for the needs and activities of seniors [10,11,12,13].

In view of the above, urban development should be analyzed in the context of identifying active aging places (AAP). These sites should be identified and assessed to ascertain the presence or absence of senior-friendly spatial solutions and to determine which parts of the city and the accompanying infrastructure need to be adapted to the needs and psychophysical abilities of seniors. The identification of AAP is a particularly important task during the COVID-19 pandemic because it will ensure the sanitary safety of seniors, prevent older adults from feeling excluded, and eliminate health threats that discourage seniors from being active. The significance of this task should be recognized by the local governments. The local authorities are responsible for implementing sustainable spatial policies that comprehensively account for the processes and phenomena that occur in a town or a city, to ensure that urban space is planned in a way that best serves the citizens and addresses their needs. Therefore, this study was undertaken to assess local governments’ preparedness for changes in the structure of society and to identify the optimal location of amenities for senior citizens.

For this reason, a new analytical approach involving Voronoi diagrams in GIS tools was proposed by the Authors to spatially identify the AAP in terms of apparent social dynamics. The developed approach is innovative and it elaborates upon preliminary research [14]. To date, the discussed topic has not been addressed by published studies or practical guides despite the fact that GIS tools are widely used for the spatial visualization of diverse phenomena, including public participation in cities [15,16] and land degradation [17]. The preliminary research conducted by the Authors [14] was the only study that relied on GIS tools, in particular Voronoi diagrams, to identify AAP. The cited authors used heat maps to graphically present AAP in the Polish city of Suwałki. The visualizations revealed the presence of ranked AAP, but the image was rather generalized within a radius of 500 m, and it provided only a general overview of seniors’ activity in the studied area. The use of Voronoi diagrams with irregular space partition enabled the visualization of the spatial distribution of AAP in a way that accounted for their actual heterogenic nature, especially in cases where data were dispersed or missing [18]. The visualization of AAP with the use of this method also facilitates analyses of selected public amenities and the relationships between these objects in the studied environment in a way that approximates human perceptions of space [19].

Voronoi diagrams are defined, and their existing applications are presented in a review of the literature in Section 2. Voronoi diagrams are used to visualize the availability of selected public amenities for seniors in an urban area in the context of the social logic of space, namely in an ordered space where individual functions and the classified AAP can be ranked [20]. This approach supports the identification of AAP in an age-friendly city in the context of the determinants of apparent social dynamics. Apparent social dynamics denote the apparent behavior of groups of people under specific circumstances at a given time, as well as the relations between these behaviors. The examined behaviors are apparent rather than direct because this study did not examine one group of people within a certain period, but different age groups living in the same place at the same time. These two determinants require the use of AAP visualization tools that illustrate the dynamics of seniors’ activity in a functionally ordered space. Voronoi diagrams fulfill these requirements. The application of Voronoi diagrams was preceded by a survey of seniors in the Polish city of Olsztyn to identify the most popular types of activities in specific age groups. In view of the above, the following research hypotheses were formulated: seniors’ activity in relation to spatial objects changes with age in the context of apparent social dynamics; the social logic of space in the analyzed area determines the accessibility of spatial objects in the context of active aging.

## 2. Literature Review

### 2.1. Active Aging Determinants and Active Aging Places (AAP)

Active aging places (AAP) cater to older people’s needs related to activation and active aging determinants. The active aging policy framework of the World Health Organization [9] outlines six sets of determinants that impact active aging across the life span, which are considered to be particularly relevant to older people as they age. These are:Determinants related to health and social services systems (health promotion and disease prevention, curative services, long-term care, mental health services);Behavioral determinants (healthy living, such as engagement in physical activity, healthy eating, oral health, appropriate medication use, and avoidance of smoking and excessive alcohol intake);Determinants related to personal characteristics (biological, genetic, and psychological factors);Determinants related to the physical environment (living in safe environments, safe housing, few environmental hazards, and environmental cleanliness);Determinants related to the social environment (sufficient social support, education and literacy, and freedom from violence and abuse);Determinants related to economic conditions (sufficient income, social protection, and opportunities for dignified work).

These six determinants correspond to aging markers (Figure 1).

A number of studies have elaborated on the World Health Organization’s active aging policy framework by promoting smart growth and active aging in communities [22,23,24]. Other authors stressed the importance of active aging in public open spaces in the city [25]. Some studies focused on environmental factors and their impact on active aging [26,27]. All of the studied determinants should be taken into account in the process of identifying AAPs which are the key spatial element of an aging-friendly city but not only in the city. A review of the literature indicates that some studies highlight the significance of space and location in the process of promoting active aging in the countryside [28,29]. The health threats associated with the COVID-19 pandemic had a negative impact on active aging. The latest research indicates that older people have reduced their activity to a minimum [30,31,32]. From a psychological and medical point of view, healthy aging is active aging [33,34]. For this reason, AAP should be identified in order to adapt them to seniors’ current needs, increase sanitary safety, minimize anxiety, and restore older people’s willingness to age actively.

### 2.2. Active Aging Determinants and Active Aging Places (AAP)

Voronoi diagrams, also known as Voronoi tessellation, are used to analyze spatial data. In this method, a selected surface or space is divided into a set of pre-defined geometric figures that fill the analyzed area completely without leaving gaps or creating overlapping shapes [35].

According to Gold [36], apart from biology, astronomy, or mathematics, Voronoi diagrams are also a useful tool for solving numerous geographic problems, for which appropriate courses of action were not always obvious. The applicability of Voronoi diagrams in geosciences is presented in Table 1.

According to Gold [36,41], “tessellation models are fundamental to our understanding and processing of geographical space, and provide a coherent framework for understanding the «space» which we exist in”. Urbanized areas, which are a complex mixture of elements of both natural and anthropogenic origin, are an example of such problematic areas [63].

Regular tessellations that divide space into polygons with equal angles and sides of the same length [64,65] are widely used to model urban areas [40]. However, they have many limitations, particularly in the context of mapping irregularly distributed phenomena in the real world that are inherently too complex for regular grids [35]. Methods based on irregular polygons enable tessellation fitting to the studied phenomenon or area in a more flexible manner, and they support the identification of hierarchical relationships within spatial arrangements in relation to the examined attributes [35]. Voronoi diagrams belong to the group of tessellation methods.

Voronoi diagrams divide Euclidean space [19] by accounting for the relationships between points and their surroundings when they are combined into groups as a result of context generalization based on the similarity of geographical structures of the analyzed area [66]. The above approach is referred to as natural tessellation, where a larger plane and points scattered in that space are divided into non-overlapping areas in the form of polygons [67]. Each fragment of the analyzed space is assigned to the nearest set of generating points [68] in accordance with the spatial structure of the examined set [69]. Each point is assigned to one region in the form of a convex polygon (so-called Voronoi cell), to which it is situated closer than any other polygons of the diagram [70]. The above implies that inside each Voronoi cell, the distance to the generating point of a particular cell is shorter than the distance to the generating points of any other cells [71].

By forming groups with identical values of selected features [66], Voronoi diagrams determine local concentrations to obtain quantitative measures of system homogeneity [72]. As a result, the spatial aspects of the studied phenomena can be visualized in a manner that accounts for their heterogeneous character, particularly when data are dispersed or in short supply [35]. In addition, data are not over-smoothed by interpolation, and their original form is preserved during processing [48].

Methods of space division that are based on the neighborhood of the analyzed points determined with Voronoi diagrams reflect the role of individual points in the analyzed set and neighborhood perceptions in a more effective way than other definitions [73]. According to Ai et al. [19], “the neighborhood relations defined by Voronoi diagrams are more closely related to human perception than other related data models”.

More importantly, Voronoi diagrams preserve the features observed in reality [53], especially information about hierarchical relations and spatial structures. The obtained results reflect the geographic features and social implications of the analyzed points, thereby meeting multiscale requirements for location expression in digital earth systems [74]. What is more, on the basis of the previous research [18], Voronoi diagrams provide a much faster product while computing the data, because they result in vector and not raster visualizations.

For the above reasons, Voronoi diagrams were selected as a research method in this study.

## 3. Materials and Methods

An empirical study was conducted with the use of qualitative (literature review, questionnaire survey, AAP classification, visualization of AAP location with the use of GIS tools) and quantitative methods (AAP ranking based on a statistical analysis of survey responses). The international and Polish literature was analyzed in the context of active aging places and the possibilities offered by Voronoi diagrams in spatial analyses.

To identify the needs of both senior citizens and persons over 55 years who will become senior citizens in the next 5–10 years, the present study was divided into two stages to determine the extent to which urban spaces are prepared to accommodate social change. The first stage involved a questionnaire survey conducted on a representative sample of 300 respondents aged over 55 years residing in the Polish city of Olsztyn (population of 171,979 in 2020). The following information was gathered during the survey: current residence, future residential preferences, architectural restraints, current trends (senior residential estates), and the choice of determinants that are most relevant for senior citizens in this aspect. The size of the population sample was determined based on analysis of statistical data and other studies involving seniors. A study investigating the housing preferences of 204 seniors and pre-seniors in the Polish city of Poznań (population of 534,813 in 2020) produced statistically significant results [75]. The questionnaire was developed based on the World Health Organization classification of age structure [5] (Table 2).

The study was carried out in the city of Olsztyn which, similarly to other urban agglomerations in Poland, is confronted with an increase in the population of older people. The study was conducted in January 2021.

In the second stage of the study, Voronoi diagrams were used to visualize the availability of selected public amenities for senior citizens in Olsztyn. The purpose of this part was to identify the existing sites or the absence of such amenities in selected urban areas in the context of investment decisions relating to the residential needs of the analyzed population. The procedure of data preparation and analysis is presented in Figure 2.

Voronoi diagrams support the determination of a region’s proximity to a selected set of points, facilitate the identification of the closest public amenities, and analyzes their accessibility within the city area. Data for the analysis, including the boundaries of the analyzed areas and the location of selected public amenities, were obtained from the OpenStreetMap service in *.SHP format.

### 3.1. Study Area—Country and City

Poland belongs to a group of European countries in the aging demographic phase, which means that the proportion of seniors (over 65 years) in the total population is increasing, whereas the proportion of younger citizens is decreasing. In the Polish statistical system, old age begins at the age of 65 years minimum (retirement age for men) or 60 years (retirement age for women). Therefore, the following classification of age structure was used in this study: 55–59, 60–74, 75–84, and over 85 years. According to Statistics Poland [76], citizens aged 65+ years will account for more than 23% of the population by 2030 (most seniors will live in cities). In 1990, a mere 13.9% of Poland’s population were aged 65 years and more. According to the demographers, population aging is inevitable. Life expectancy will most likely continue to increase, which will increase both the total number and the percentage of elderly citizens. The above trend has been observed in most Polish cities in recent decades (Table 3). The present study was conducted in Olsztyn, the capital city of the Voivodship of Warmia and Mazury in north-eastern Poland (Figure 3).

Statistics Poland [76] data show changes characteristic of the Polish population in recent years and point to an increase in the elderly population. According to Eurostat forecasts [77], life expectancy at the age of 65 is 20 years for Polish women and 16 years for Polish men. Aging processes are clearly visible at the national level, including in the studied city of Olsztyn.

Olsztyn has an area of 88.33 km^2^. The city is intersected by four rivers, and it features 15 lakes that occupy 9% of the city’s area. Olsztyn is also characterized by very high proportions of woodlands and urban greenery (including two nature reserves: Mszar and Redykajny) that cover 58% of the city’s area (Figure 4).

### 3.2. Preparation of the Survey Questionnaire

In order to determine the apparent social dynamics of older people, seniors were divided into the age groups proposed by the World Health Organization [5]. The last two levels of the World Health Organization classification were adjusted to the statistical classification proposed by Statistics Poland [76] because people older than 85 years account for only 3.9% of the Polish population. The social dynamics of this age group should be examined in view of the frequently validated observation that sensory processing abilities and mobility decline with age. The survey questionnaire was prepared based on an extensive review of the literature on active aging and the associated barriers. Special emphasis was placed on the World Health Organization [78,79,80] recommendations for aging and health. According to the research conducted by Statistics Poland [76], vision is the first sense that is affected by age (in spite of using glasses, more than half of the elderly people in Poland have problems with their sight), followed by hearing and independent walking. The ability to independent self-care declines with age. Almost 45% of elderly people (65+ years) need help with daily tasks such as shopping, light housework, or taking care of financial issues. Similar observations were made by Lee and Dean [81] in a study analyzing older people’s mobility in cities, by Bertocchi et al. [82] in a research paper on investments, and by Cocco et al. [83], Yogo [84], Nakajima and Telyukova [85], and Angelini et al. [86] in studies exploring seniors’ housing choices and changes in the place of residence. Seniors feel the need to live in their place of residence for as long as possible because the local environment is conducive to physical and mental health and ensures well-being [87,88,89].

On the basis of obtained information, the preliminary stage of the preparation of the survey was to develop a local (Polish) and a global (general) profile of a senior. This crucial step in the methodology provided a way to verify the types of senior citizens’ preferences for activity by confronting it with the survey’s results. The profile consisted of eight categories regarding (Table 4): demography (Table 5), psychology (Table 6), behavior (Table 7), economics (Table 8), activity needs (Table 9), proximate space (Table 10), transitional space (Table 11), and future (Table 12).

Based on the results of the above research and taking into account the developed senior profile, nine groups of activities were described in the questionnaire (Culture and science, Physical activity/Sports, Recreation, Health, Administration, Transport, Worship, Home activity, Shopping). The questionnaire contained eight questions aiming to determine the locations where seniors undertake different types of activities, the frequency of these activities, and the means of reaching these locations. The questionnaire contained mostly closed-ended and single-choice questions. The first two questions concerned gender and age within the defined age categories (55–59, 60–74, 75–84, and over 85 years). In the third question, the respondents were asked to indicate their place of residence in the city of Olsztyn. In the fourth question, seniors were asked to indicate the most frequent mode of travel to their sites of daily activity: on foot, two-wheeled vehicle, four-wheeled vehicle, or rail vehicle. Question five concerned the preferred means of transport, including taxi, public transport (bus, tram, train), bicycle, scooter, moped/motorcycle. The purpose of the sixth question was to determine the optimal distance to various activity areas that an older person is able to reach on foot in the following ranges: less than 500 m, 500–1000 m, 1000–1500 m [102]. The seventh question concerned seniors’ independence in everyday activities. The last question was an expanded single-choice table of spatial objects that were frequented by seniors and correlated with the frequency of visits. The table contained nine groups and places of activity and their locations. The frequency of a given activity was used as a basis to calculate the weights for different types of activities. The results were used to identify factors that prevented the respondents from engaging in a given type of activity (insufficient funds, health problems, excessive distance, no need).

The developed questionnaire was largely universal, but the results should be generated and averaged locally because senior profiles are unique in a given space (local functional and spatial conditions, facilities, and policies).

## 4. Results

### 4.1. Questionnaire Survey

The results of the survey conducted on a representative sample of 300 seniors were used to rank the locations of senior activity in the context of apparent social dynamics (Table 13).

Women accounted for 56%, and men accounted for 44% of the studied population. The largest group (113 persons, 38%) were seniors aged 60–75 years. The remaining age groups were persons aged 55–59 years (108 respondents, 36%), 76–84 years (54 respondents, 18%), and persons older than 85 years (26 respondents, 8%). The results were considered reliable because they were correlated with the age structure of senior inhabitants of Olsztyn (according to Table 3). The third question concerning the means by which seniors reach the locations of everyday activities confirmed the “aging in place” theory [103] which states that seniors most often stay in their place of residence/neighborhood. The survey demonstrated that 87% of seniors aged 60 to over 85 years travel on foot. These findings correspond with the answers concerning the optimum distance that a senior can cover on foot. It appears that the older the person, the lower their ability to take long walks. Most pre-seniors declared that they were able to walk a distance of over 1000 m. In the next age category, mobility decreased proportionally, which proves that older people most frequently travel within their own neighborhoods. In the pre-senior group of 55- to 59-year-olds, four-wheeled vehicles were indicated by 81% of the respondents, which suggests that the activity of this age group extends beyond their place of residence (neighborhood). The question concerning the preferred means of transport did not reveal a correlation between the choice of a specific vehicle and gender. The proportions were equal. However, such a correlation was observed in an analysis of the age structure. Most 55- to 59-year-olds had a preference for traveling in their own cars (82%), and similar results were noted in the 60–75 years age group (76%). The remaining age groups were more likely to choose public transport (77% of respondents in the 76–84 years group; 14% of respondents in the 85+ years group). None of the respondents used scooters, whereas bicycles were chosen by 27% of the respondents in the 55–59 years age group, 19% in the 60–75 years age group, and none in the 76–84 years age group (due to health problems or lack of need). Approximately 20% of all seniors choose taxis as an occasional means of transport, probably due to limited financial means. A lack of need as the reason for not undertaking any activity was indicated by 86% of the respondents. Other reasons for not undertaking any activity were health problems (10%), insufficient funds (3%), and excessive distance (1%).

The second- last question demonstrated that 89% of the respondents were independent in their daily activities. The remaining 11%, mainly seniors aged 75–85 years, were either not fully independent (8%) or completely dependent (3%). This was confirmed by the answers indicating that health problems were the main reason for not engaging in any activity.

The questionnaire survey (Table 13) revealed that the level of activities involving the identified objects increased with the frequency of these activities in the context of apparent social dynamics. In all age groups, activity levels were highest in the respondents’ place of residence, which confirms the validity of the “aging in place” hypothesis. For this reason, the accessibility of other objects of activity should be determined in relation to the place of residence. Other sites of increased activity include pharmacies, local groceries, churches, parks and squares, libraries, clinics, and health centers, specialist healthcare centers, and public transport stops.

The respondents from different age groups could choose between three options to describe the frequency of their activities: at least monthly, at least quarterly, and at least yearly. Ranks were calculated by determining the median in ordered sequences of responses. Activity levels clearly decreased with a rise in age. The calculated ranks indicated which activities and sites/objects were preferred by the surveyed seniors. The results were used to select the most important AAPs which were visualized on active aging maps with GIS tools and the Voronoi diagram method.

### 4.2. Spatial Analysis with Voronoi Diagrams

The aim of the study was to identify and examine active aging places in the selected area with the use of Voronoi diagrams as a method supporting irregular division of space. To accomplish this task, the acquired spatial data, including the results of the questionnaire survey conducted on senior residents of Olsztyn, were analyzed with the use of QGIS and ArcGIS Pro Software. According to Siejka [45], Voronoi diagrams and GIS tools can be successfully applied to visualize spatial information regarding the number and distribution of the studied objects in space and to conduct preliminary analyses of the attributes that determine the diversity of the examined objects.

Maps depicting the spatial distribution of selected public amenities in Olsztyn were generated in the first stage of the study. The information about the location of the studied objects was obtained from the OpenStreetMap service in the form of shapefiles (*.SHP), and the acquired data were validated based on real-world data. Missing objects were added to the database by indicating their position on maps. The resulting visualization demonstrated that selected objects were unevenly distributed across the studied area, which could be attributed to the complex spatial structure of Olsztyn and the presence of woodlands, water reservoirs, and industrial areas (Figure 4).

In the next stage of the process, tessellation was performed by dividing Olsztyn’s area into a grid of Voronoi cells, where each generating point (denoting selected public amenities) formed its own polygon. “A Voronoi cell represents an area of influence of the data point it contains, and thus the local density in the proximity of a given point can be determined as the inverse of the cell area” [70,104,105]. This approach is used mainly to determine the accessibility of studied objects or the risk of occurrence of selected phenomena [19,38,40,46,106,107,108].

The resulting mesh of Voronoi diagrams shows the spatial distribution of public facilities, where each cell is assigned to exactly one object. The density of these cells supports the identification of clusters with a higher number of objects. The number of objects per 1 km^2^ was calculated in each cell, and then the resulting cells were aggregated into five groups (on a 5-grade scale) based on their density (Figure 5 and Figure 6).

The generated maps indicate that the largest number of the analyzed objects from each group are situated in the central and southern parts of the city. Based on these visualizations, it can be concluded that Olsztyn is characterized by high accessibility to most of the examined public facilities, including public transport stops and grocery shops, but there is relatively low availability of pharmacies, health care facilities, and places of worship.

In the next step, each type of facility was compared with the location of city districts inhabited by most of the surveyed senior citizens. The procedure is shown in the example of pharmacies in Figure 7.

The comparison revealed that most of the identified areas are characterized by a high density of pharmacies (at least one per 1 km^2^), which implies that senior citizens living in these districts have relatively easier access to pharmacies than seniors residing in other parts of the city.

Seniors’ walkability and the apparent dynamics of seniors’ activity in different age categories (refer to Section 3.2) were considered in the second stage of the study. Seniors in the 60–75 years age group are generally able to walk a distance of around 1500 m; seniors aged 76–84 years—1000 m, and seniors aged 85 years and older—500 m. The results were visualized by drawing radiuses around the locations of the analyzed activity sites. An example for pharmacies is presented in Figure 8.

In the following step, seniors’ activity was compared with the use of Voronoi diagrams visualizing the density of the studied objects (Figure 9).

An analysis of seniors’ apparent dynamics demonstrated that as older people’s walkability decreases over time as they get older, so does their activity. The apparent dynamics of active aging places were analyzed by examining the location of districts inhabited by most of the surveyed seniors in Olsztyn. The results are presented in Figure 10 on the example of pharmacies.

Maps depicting the accessibility of the studied AAP were generated in the last stage of the study. A map illustrating the accessibility to pharmacies is presented in Figure 11. It can be assumed that pharmacies located within a distance of 500 m from the place of a residence constitute the highest or first rank in the social logic of space. Sites that are accessible within a radius of 500–1000 m occupy the second rank in the urban space, and sites accessible within a radius of 1000–1500 m occupy the third rank. This approach can be adopted to rank the urban area in the context of other AAPs. It can thus be assumed that the youngest seniors are in a privileged situation because most of the neighborhoods have good access to the studied public facilities. Neighborhoods located outside the accessibility radiuses shown on the map (Figure 11) represent urban areas that require additional investment in the future (especially in the context of active aging).

## 5. Discussion

In the described approach, Voronoi diagrams were used to generate maps of AAPs and to determine the accessibility of selected sites in space. The results were used to rank the urban space in the context of AAP, and to identify the studied sites in view of the social logic of space and apparent social dynamic. Therefore, the results of the study validate the research hypotheses postulating that: (1) Seniors’ activity in relation to spatial objects changes with age in the context of apparent social dynamics; (2) The social logic of space in the analyzed area determines the accessibility of spatial objects in the context of active aging.

Heat maps, which are generally used in research studies of this type, depict only the presence and density of the analyzed facilities in the selected area. The proposed method organizes the geometry of the studied space and indicates the extent to which the examined urban space meets senior citizens’ demand for specific public facilities in the context of their activity and active aging. The research objective was achieved based on the assumption regarding the apparent social dynamics of older people.

The visualization of AAP with the use of Voronoi diagrams facilitates the presentation of public utilities that are unevenly distributed in irregular space in a way that approximates human perceptions of the space where seniors live and move. As a result, this approach can be used to illustrate the spatial relationships between the examined AAP in the vicinity of seniors’ places of residence.

However, the proposed approach has certain limitations. The developed method is effective only in analyses of individual facilities, and the clarity of the generated visualizations decreases considerably when numerous sites are studied. The above implies that the proposed method organizes space only in the context of a single selected facility.

## 6. Conclusions

The results obtained under the study extend the existing body of knowledge on the possibilities of application of Voronoi diagrams in socio-economic and geographical studies (i.e., Refs. [19,38,45,46,108,109,110,111]). The information generated with the use of the developed approach can be used by the local authorities to analyze the demand for specific facilities in selected areas. The Accessibility Plus Program financed by the European Funds [112] is one of the potential areas of application of the developed method.

According to Masuyama [113] and Figurska and Bełej [72], accessibility maps can enable and facilitate the decision-making processes in the search for new accommodation for seniors who want to change their current place of residence. In this context, the developed method not only supports urban spatial policies that are consistent with the concept of the social logic of space by promoting age-friendly cities but it also offers a new methodological solution for delimiting areas that are in need of revitalization in the context of visualizing other phenomena in urban space. The main strength of the proposed approach is that it is universal and can be applied to various spatial conditions and different types of phenomena or objects. The only requirement is adequate and sufficient access to data.

## Figures and Tables

**Figure 1 ijerph-19-07447-f001:**
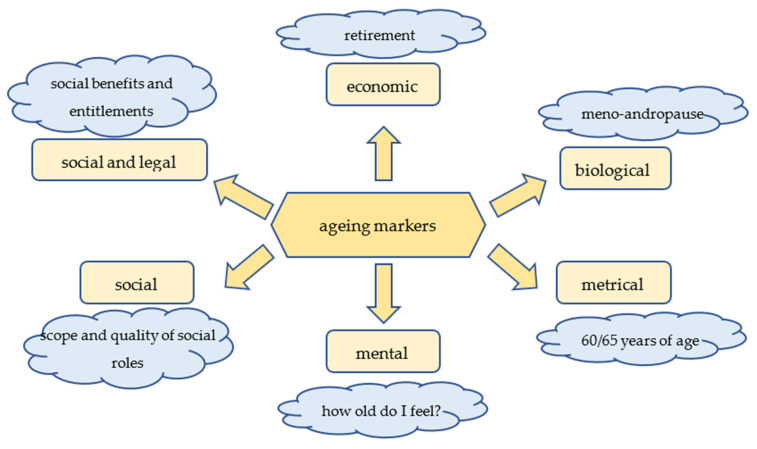
Aging markers. Source: own elaboration based on [21].

**Figure 2 ijerph-19-07447-f002:**
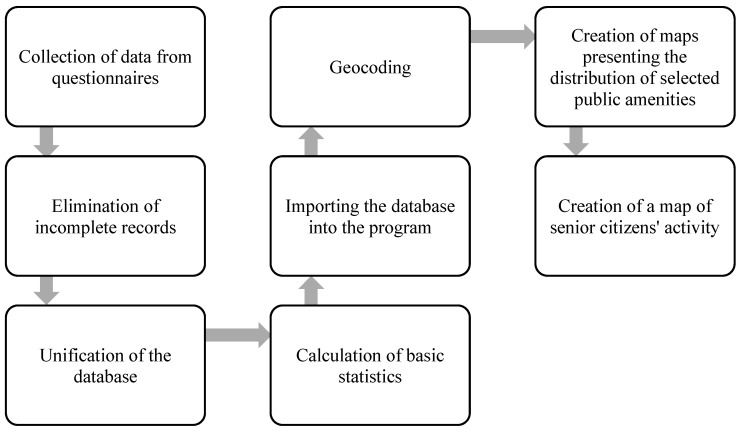
Procedure of preparing and analyzing the data for Olsztyn. Source: own elaboration.

**Figure 3 ijerph-19-07447-f003:**
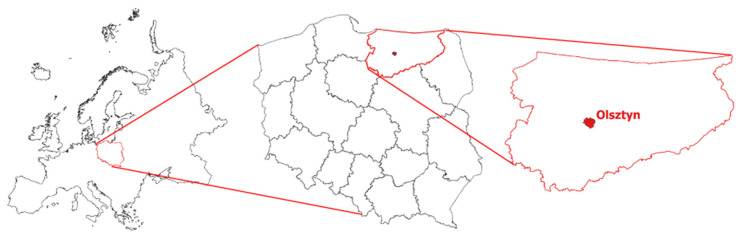
Location of the Voivodship of Warmia and Mazury in Poland (**left**) and Olsztyn in the Voivodship of Warmia and Mazury (**right**). Source: own elaboration.

**Figure 4 ijerph-19-07447-f004:**
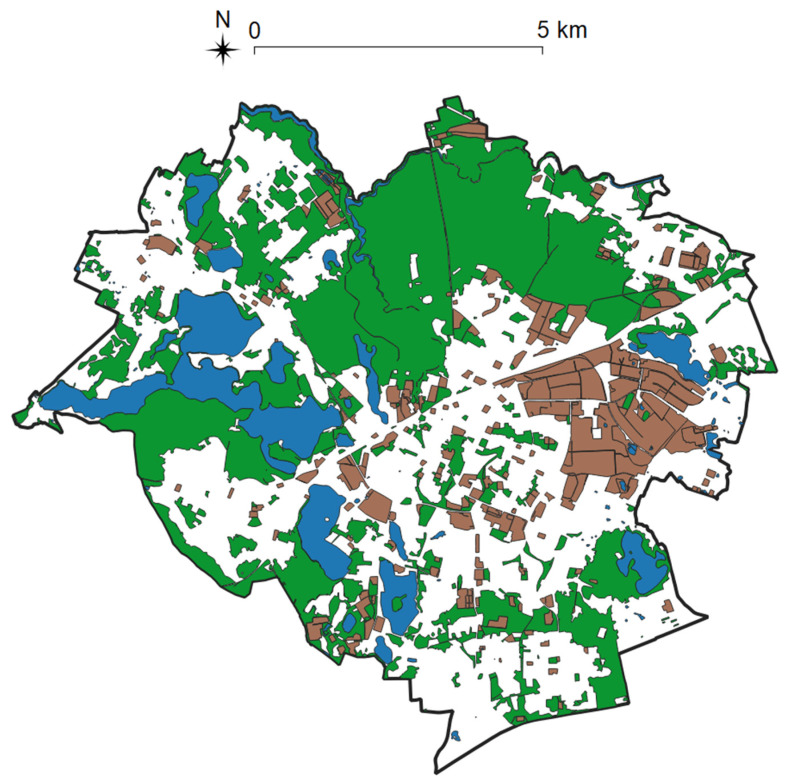
Location of woodlands (green), water reservoirs (blue), and industrial areas (brown) in Olsztyn. Source: own elaboration.

**Figure 5 ijerph-19-07447-f005:**
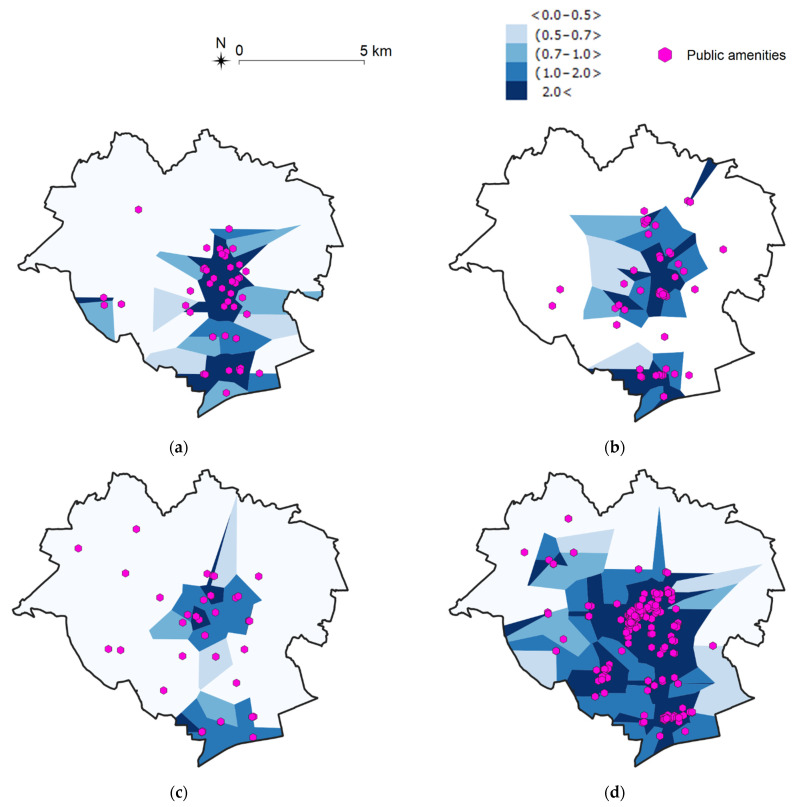
Spatial accessibility of selected public amenities in Olsztyn [units/km^2^], part 1. Source: own elaboration. (**a**) Pharmacies; (**b**) Health care facilities; (**c**) Churches; (**d**) Restaurants.

**Figure 6 ijerph-19-07447-f006:**
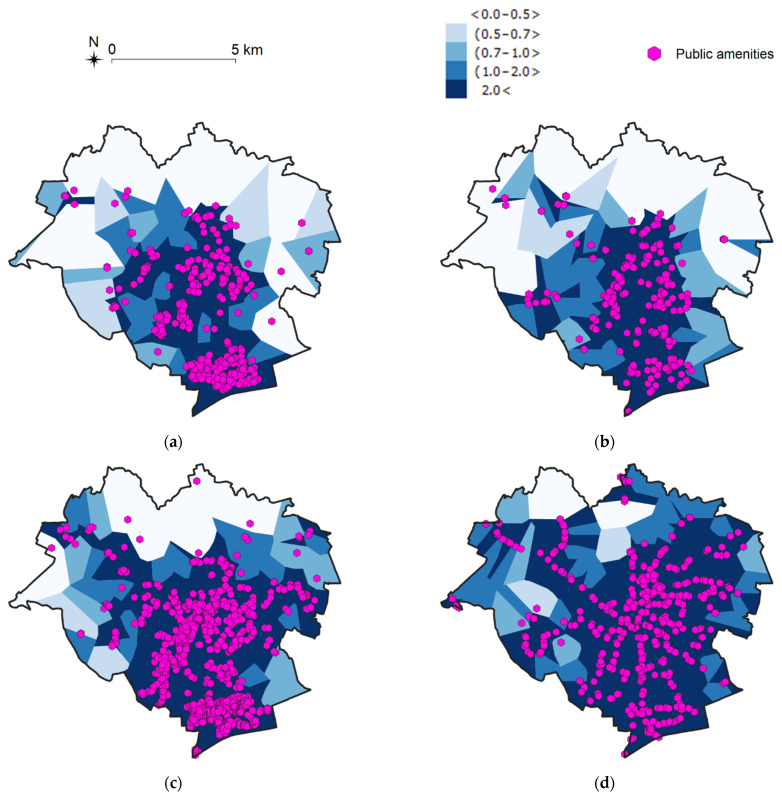
Spatial accessibility of selected public amenities in Olsztyn [units/km^2^], part 2. Source: own elaboration. (**a**) Sports facilities; (**b**) Grocery stores; (**c**) Parking lots; (**d**) Public transport stops.

**Figure 7 ijerph-19-07447-f007:**
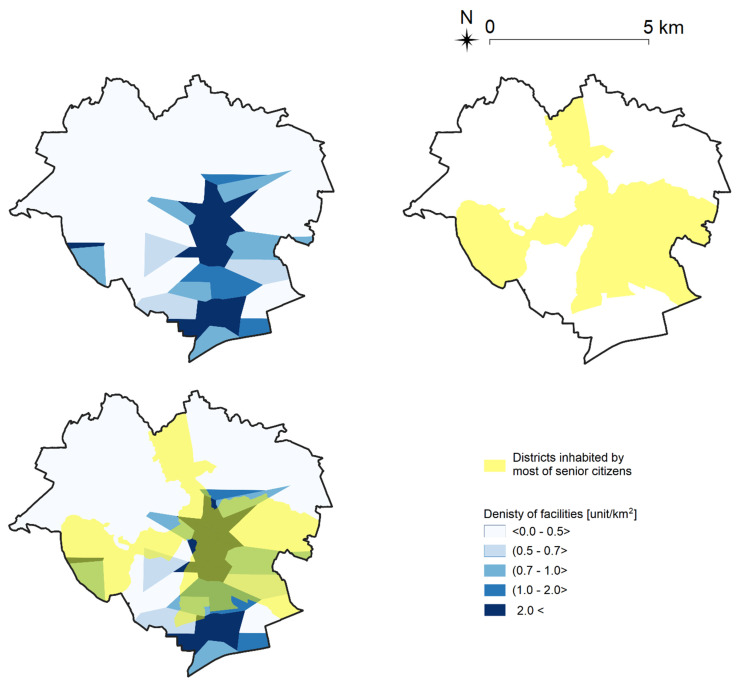
Comparison of the density of pharmacies (blue) and the location of districts inhabited by most of the surveyed senior citizens (yellow) in Olsztyn. Source: own elaboration.

**Figure 8 ijerph-19-07447-f008:**
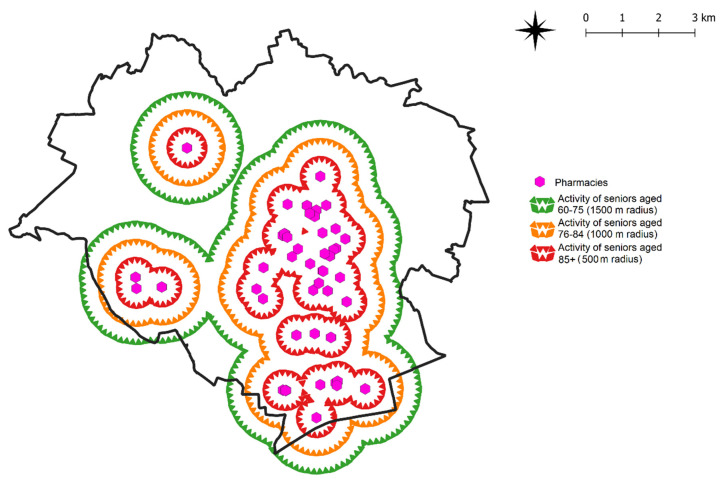
Senior activity sites in Olsztyn on the example of pharmacies. Source: own elaboration.

**Figure 9 ijerph-19-07447-f009:**
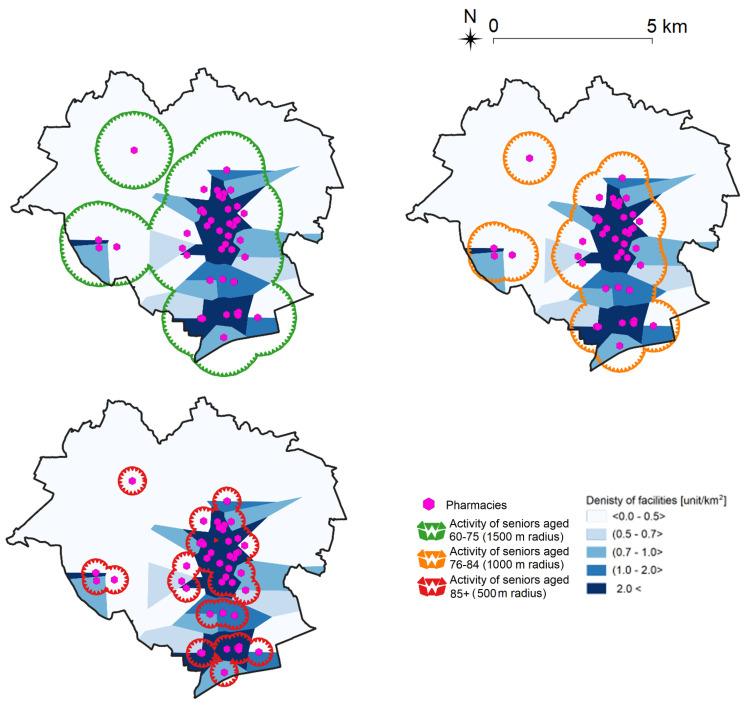
Apparent social dynamics of seniors’ activity on the example of pharmacies. Source: own elaboration.

**Figure 10 ijerph-19-07447-f010:**
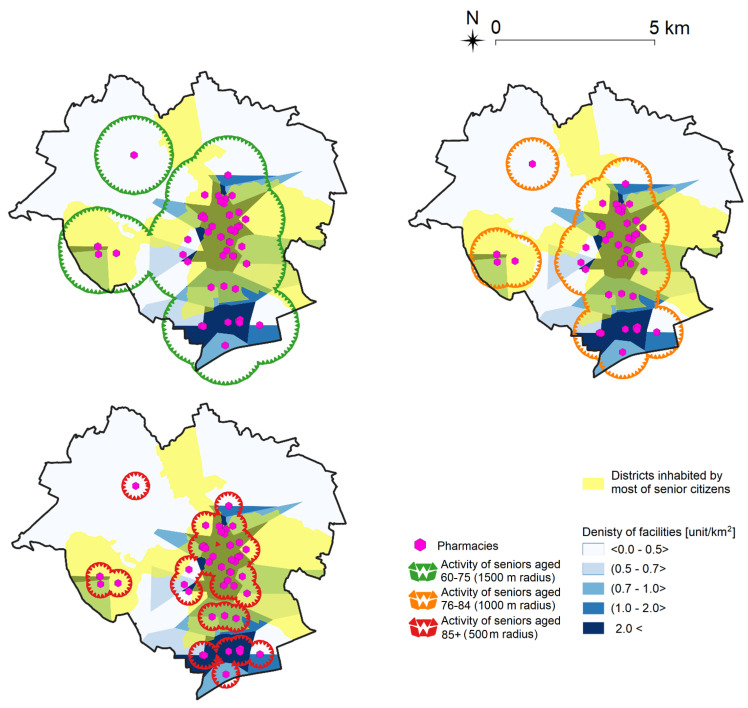
Apparent social dynamics of seniors’ activity on the example of pharmacies and the location of districts inhabited by most of the surveyed seniors in Olsztyn. Source: own elaboration.

**Figure 11 ijerph-19-07447-f011:**
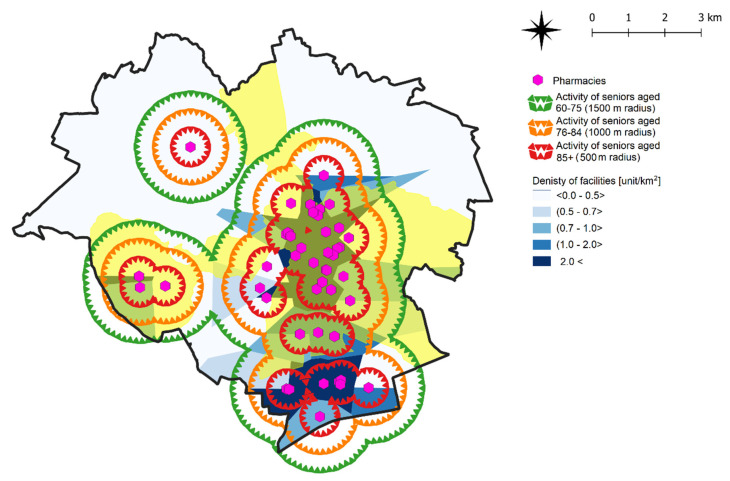
Apparent social dynamics of seniors’ activity on the example of pharmacies and the location of districts inhabited by the surveyed seniors in Olsztyn, for all age groups. Source: own elaboration.

**Table 1 ijerph-19-07447-t001:** Applicability of Voronoi diagrams in geosciences. Source: own elaboration.

Scientific Field	Use
Geodesy, geography, gravimetry	Analysis of traffic accidents and crime [37,38,39]; Analysis of roadmaps and urban networks [40]; Analysis of the total impact of infrastructure elements on housing distribution [40]; Atmospheric simulations [41]; Dynamic simulations of space [42,43]; Dynamic visualizations on maps [44]; Evaluation of real estate market activity [45]; Generation of real estate ownership maps [46]; GPS precise positioning [47]; Gravity and gravimetric surveys [48]; Indexing of coordinates [36]; Modelling of consumer behaviors [49]; Modelling retail areas [42,46,50]; Planning of the shortest route [19,49]; Solving problems with location optimization [42,51,52]; Spatial analyses and simulations [53].
Forestry, agriculture	Analysis of woodworm attacks [54]; Generation of forest stand maps [41]; Survey of weed growth in agricultural fields [55].
Geology, seismology, hydrology	Analysis of the distribution and force of earthquakes [42,56,57,58,59,60,61]; Analysis of tectonic plates [61]; Modelling oil deposits [42]; Analysis of precipitation rates [36,50,62].

**Table 2 ijerph-19-07447-t002:** Age structure of older people. Source: own elaboration based on [5].

Structure of Older People	Years	Description
pre-old age	55–59	pre-senior
early old age	60/65–74	young old
old age proper	75–89	old-old
oldest-old	90+	oldest-old-lifelong

**Table 3 ijerph-19-07447-t003:** Demographic data for Poland and Olsztyn. Source: own elaboration based on [76].

Demographic Data	Poland	Olsztyn
Population	38,265,013	171,249
Older people (aged 60 years and more)	26%	27%
Percentage of women among older people	58%	60%
Growth rate of the older population		
1990 (urban areas) 2020 (urban areas)	10.2% 25.6%	9.0% 27.2%
Old-age dependency ratio (number of persons of non-working age per 100 persons of working-age)	46	47
AGE STRUCTURE OF SENIORS IN CITIES WITH A POPULATION OF 100,000:		
55–59 60–75 76–84 85 and older	10% 11% 4% 4%	6% 20% 5% 2%

**Table 4 ijerph-19-07447-t004:** Senior’s profile—categories. Source: own elaboration on the basis of [90,91,92,93,94,95,96,97,98,99,100,101].

No.	Profile Categories	Description
1	Demographic profile	A set of data about a particular social group (senior citizens), including: age, gender; ethnic group, marital status and number of children (fertility), geographical location; occupation; education;
2	Psychological profile	A set of data based on a demographic database and psychometrics, understood as a collection of data about senior citizens, focused on their life attitudes, aspirations, interests, lifestyles, etc. It provides a deeper and more subjective understanding of who senior citizens are and how they think. It covers issues such as: attitude towards the world, aspirations and dreams, interests, personality, priorities, problems and concerns.
3	Behavioral profile	A set of data building on the previous two profiles indicating the specific behavior of senior citizens under study. It sets the parameters for a positive or negative attitude towards the world and life, determining a space in which one feels comfortable.
4	Economic profile	A set of data identifying the financial capabilities (income) and expenditures carried out by households of senior citizens.
5	Profile of activity needs	A set of data identifying the needs of senior citizens in terms of their expectations from the space in which they live, spend their free time, and become active.
6	Profile of the proximate space	A set of data concerning the place of residence of senior citizens (their homes).
7	Profile of transitional space	A set of data concerning the surroundings of the place of residence of senior citizens (neighborhood they live in).
8	Profile of the future	A set of data about the possibility of relocation (moving to another place) by senior citizens.

**Table 5 ijerph-19-07447-t005:** Demographic profile of a Polish senior. Source: own elaboration on the basis of [90,91,92,93,94,95,96,97,98,99,100,101].

Data	
Population age structure in households of senior citizens:	
under 14 years old:	1.8%
15–34:	4.4%
35–54:	6.9%
55–59:	3.2%
60–64:	15.0%
65 years and over:	68.5%
Gender:	
males:	41.9%
females:	58.1%
Ethnic group:	
Polish:	96.8%
non-Polish:	4.4%
Marital status and number of children (fertility):	
single:	6.5%
married:	65.1%
widowed:	22.3%
divorced:	6.1%
Old age for men mostly means living with another person, and for women it is mostly being alone.	
Geographical location (voivodeship):	
Mazowieckie:	13.7%
Sląskie:	12.5%
Wielkopolskie:	8.6%
Małopolskie:	8.4%
Dolnośląskie:	7.8%
Iódzkie:	6.8%
Pomorskie:	5.8%
Lubelskie:	5.6%
Kujawsko-Pomorskie:	5.4%
Podkarpackie:	5.3%
Zachodniopomorskie:	4.6%
Warmińsko-Mazurskie:	3.6%
Swiętokrzyskie:	3.4%
Podlaskie:	3.1%
Opolskie:	2.7%
Lubuskie:	2.6%
Occupation:	
employees:	16.4%
self-employed (including family workers):	6.3%
unemployed:	3.1%
retirees:	56.6%
pensioners (due to disability), incapable of working due to their Health status:	6.9%
persons keeping a household, taking care of other persons:	1.5%
other inactive:	7.8%
Education:	
higher/tertiary:	16.2%
post-secondary:	2.4%
upper secondary vocational:	24.3%
upper secondary general:	9.2%
secondary:	33.5%
elementary/basic vocational:	29.1%
lower secondary:	0.8%
completed primary:	17.3%
no school/formal education:	0.7%
Older people have on average a lower level of education than the younger generation. There remains a wide disparity in educational attainment between seniors living in urban and rural areas.	

**Table 6 ijerph-19-07447-t006:** Psychological profile of a Polish senior. Source: own elaboration on the basis of [90,91,92,93,94,95,96,97,98,99,100,101].

Data	
Satisfaction with:	
their current family situation:	66.3%
relationships with other people:	77.4%
their current financial situation:	31.8%
their material living conditions:	52.9%
the amount of time available:	82.9%
ways of spending free time:	65.0%
their health:	23.5%
Senior citizens: spend time with their grandchildren, often replacing a babysitter; love working in the garden and on the allotment; consider family as a priority; move within the city limits; do not like to travel far; spend time actively walking; like to be with people of their age; like to cook; attend religious services at least a few times a month.	

**Table 7 ijerph-19-07447-t007:** Behavioral profile of a Polish senior. Source: own elaboration on the basis of [90,91,92,93,94,95,96,97,98,99,100,101].

Data	
Active citizenship and voluntary activities:	
participation in informal voluntary activities:	40.5%
participation in formal voluntary work:	9.8%
participation in activities related to active citizenship:	5.7%
Relations with other people:	
having friends:	61.2%
relying on someone’s help:	74.5%
feeling safe in the neighborhood:	80.4%

Older people face challenges in redefining their social roles as they exit the labor market and enter retirement, but also diminishing opportunities due to increasing constraints resulting from declining health and gradually increasing functional limitations. Seniors are family active, enjoy spending time in nature and with their families.

**Table 8 ijerph-19-07447-t008:** Economic profile of a Polish senior. Source: own elaboration on the basis of [90,91,92,93,94,95,96,97,98,99,100,101].

Data	
Financial capabilities:	
average monthly income [PLN]:	1940.87
average retirement pay [PLN]:	1633.43
average monthly expenditures [PLN]:	1335.94
at-risk-of-poverty threshold for a single person [PLN]:	20,685.00
Evaluation of material situation:	
very good:	17.6%
rather good:	22.6%
average:	51.7%
rather bad:	6.7%
bad:	1.4%
“Making ends meet”:	
with great difficulty:	6.9%
with difficulty:	16.0%
with some difficulty:	38.1%
fairly easily:	29.7%
easily:	7.4%
very easily:	1.9%

**Table 9 ijerph-19-07447-t009:** Profile of activity needs of a Polish senior. Source: own elaboration on the basis of [90,91,92,93,94,95,96,97,98,99,100,101].

Data	
Activities:	
attendance at the cinema (at least once a year):	13.7%
attendance at a live performances (at least once a year):	14.5%
visits to cultural sites (at least once a year):	24.0%
attendance at live sports events (at least once a year):	11.2%
going to the library or reading room (at least once a month):	6.6%
going to discos or dances (at least once a month):	1.0%
sporting activity (at least once a month):	9.1%
going for walks, spending time outdoors (e.g., in the garden) (at least once a month):	67.5%
listening to music (at least once a month):	28.9%
watching movies (at least once a month):	9.9%
spending time on personal hobbies (at least once a month):	18.8%
getting together with their family (at least once a week):	20.7%
contact with family (at least once a week):	30.1%
getting together with their friends (at least once a week):	19.5%
contact with friends (at least once a week):	27.2%
using social media (daily):	1.9%
practicing artistic activities (at least once a week):	8.4%
reading daily newspapers (daily)	29.2%
holidays, excursions, rallies abroad:	12.9%
holidays, excursions, rallies domestically:	39.0%
visiting family, relatives, friends abroad:	12.4%
visiting family, relatives, friends domestically:	38.4%
visiting allotments:	22.8%

**Table 10 ijerph-19-07447-t010:** Profile of the proximate space of a Polish senior. Source: own elaboration on the basis of [90,91,92,93,94,95,96,97,98,99,100,101].

Data	
Place of residence:	
detached house:	39.5%
semi-detached house:	4.8%
apartment:	55.5%
average usable floor space by 1 person [m^2^]:	2.9

**Table 11 ijerph-19-07447-t011:** Profile of transitional space of a Polish senior. Source: own elaboration on the basis of [90,91,92,93,94,95,96,97,98,99,100,101].

Data
Seniors appreciate where they live, mostly because of the possibilities for good communication within their towns. In addition, access to green space parks is extremely important to them (34%). In their surroundings seniors would like to: be close to family 39%; be close to friends 5%; live in a quiet neighborhood 25%.

**Table 12 ijerph-19-07447-t012:** Profile of the future of a Polish senior. Source: own elaboration on the basis of [90,91,92,93,94,95,96,97,98,99,100,101].

Data
90% of seniors would not change their place of residence.

**Table 13 ijerph-19-07447-t013:** Active aging groups and places in the city. Source: own elaboration based on [14].

No.	Active Aging Groups	Active Aging Places		Significance of Active Aging Places in the Aging Structure			
		Age		55–59	60–75	76–84	85+
1	Culture and science	Cinema		1	1	0	0
		Theatre		0	1	0	0
		Museum/Gallery		0	0	0	0
		Library		2	2	2	1
		Further education sites (language courses, postgraduate studies)		0	0	0	0
		Concerts/Festivals		1	1	0	0
		Visiting relatives (family, friends, neighbours)		3	3	2	2
2	Physical activity/Sports	Swimming pool		2	2	0	0
		Gym		0	0	0	0
		Nordic walking sites		2	0	0	0
		Parks and squares/Walks		3	3	0	0
		Cycle paths		0	1	0	0
3	Recreations	Travel agencies/Sightseeing tours		0	1	0	0
		Dancing clubs/Dancing events		0	0	0	0
		Public sites for playing Chess/Cards		0	0	0	0
		Cafés/Restaurants		1	0	0	0
		Parks/Botanical gardens		2	2	0	0
		Allotments gardens (garden work)		2	2	0	0
4	Health	Clinics and health centres		2	0	2	2
		Specialist healthcare centres		2	0	2	2
		Hospitals		0	0	0	0
		Sanatoria		0	0	0	0
		Rehabilitation centres		0	0	0	0
5	Administration	Town hall		1	1	0	0
		Social welfare office		0	0	0	1
		Property manager		0	0	0	0
		Bank (paying the bills)		0	1	2	0
		Post office		1	1	2	0
		Police station		0	0	0	0
		Tax office		0	0	0	0
		Social Insurance Institution		0	0	0	0
6	Transport	Bus stops—Public transport		3	0	1	0
		Garage space/Own car		3	2	0	0
		Cycling path		0	0	0	0
		Taxi stands		0	0	0	0
7	Worship	Churches		2	3	3	0
		Other objects and places of worship		0	0	0	0
		Cemeteries		2	2	2	1
8	Home activity	Home space	Watching TV	3	3	3	3
			Reading	3	3	3	0
			Cleaning	3	3	3	1
			Entertaining	3	3	3	3
9	Shopping	Supermarkets		3	3	0	0
		Local groceries		3	3	2	0
		Shops with a specialized product range, i.e., pharmacies		2	3	2	0
		Internet		2	2	0	0

## Data Availability

Not applicable.

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
