# Peer review of "Voronoi Diagrams for Senior-Friendly Cities"

_ijerph, 2022, doi:10.3390/ijerph19127447_

Round 1
Reviewer 1 Report
Dear Authors,
Thank you for the opportunity to review this article. The Authors presented an interesting look at the following research problem. The title of the study "Voronoi diagrams for senior-friendly cities" corresponds to its content.
The article contains small errors which, after removing and introducing additional content, will enrich the scientific work.
They are:
1. The bibliography is not formatted in accordance with the requirements of the MDPI publishing journals. For example: In the article is written (Romshoo et al., 2020) should be [No.] Please correct it.
2. The Reference contains 105 items, but there are many more in the text of the article. Not all of them are mentioned in the reference (line: 71-90).
3. "Authors" - Name not cited: line 71, 75.
4. In this article, figure 3 is repeated twice ( line: 249 and 254).
5. Figure 3 please complete with a more detailed graphic representation. Perhaps it is worth highlighting the research object against the background of Europe. It is proposed to place Poland against the background of Europe. Please correct it.
6. Table 1 also includes citation errors (Cua, 2004; G. Cua & Heaton, 2007; Dong, 2008; Grasso et al., 2007; Iervolino et al., 2007; Satriano et al., 2007; Schoenberg et al., 2009). It should be [No.] Please correct it.
7. In the 5. Discussion and conclusions, more references to world literature are proposed. Please complete this and the discussion will be more interesting and valuable from a scientific point of view.
It should be noted that the whole of the study is cognitive and contains important scientific elements. The article was written at a very good academic level.
In relation to the above, I express the opinion that the work submitted for review should be published in its entirety after taking into account the comments of the reviewer and does not require a review again.
Author Response
Responses to the review of the article entitled:
Voronoi diagrams for senior-friendly cities
submitted to the IJERPH
Upon the Editorial Board’s request, we have made every effort to account for the Reviewers' comments in the revision process. We hope that the article meets the expectations of the Reviewers and the Editorial Board and that it is acceptable for publication in the IJERPH in its present form.
The Authors would like to thank the Reviewers for the time and effort invested in the revision process. The Reviewers’ insightful comments have significantly improved the manuscript’s potential to advance scientific knowledge. We hope that the revised manuscript is suitable for publication.
The responses to the Reviewers’ comments are marked in red. A version of the manuscript in the track changes mode is attached to the resubmission.
Best regards,
The Authors
Response to Reviewer 1 Comments
Point 1. The bibliography is not formatted in accordance with the requirements of the MDPI publishing journals. For example: In the article is written (Romshoo et al., 2020) should be [No.] Please correct it.
Response 1: The references have been corrected. It was an obvious mistake of the reference software – thank you for the comment.
Point 2. The Reference contains 105 items, but there are many more in the text of the article. Not all of them are mentioned in the reference (line: 71-90).
Response 2: The references have been corrected. It was an obvious mistake of the reference software – thank you for the comment.
Point 3. "Authors" - Name not cited: line 71, 75.
Response 3: The authors did not want to write their names to keep the article text anonymous. But these references have already been corrected.
Point 4. In this article, figure 3 is repeated twice ( line: 249 and 254).
Response 4: Apologies for the obvious mistake. The unnecessary figure has been removed.
Point 5. Figure 3 please complete with a more detailed graphic representation. Perhaps it is worth highlighting the research object against the background of Europe. It is proposed to place Poland against the background of Europe. Please correct it.
Response 5: Location of the analysed are in Europe has been added to the figure, as the Reviewer suggested.
Point 6. Table 1 also includes citation errors (Cua, 2004; G. Cua & Heaton, 2007; Dong, 2008; Grasso et al., 2007; Iervolino et al., 2007; Satriano et al., 2007; Schoenberg et al., 2009). It should be [No.] Please correct it.
Response 6: The references have been corrected. It was an obvious mistake of the reference software – thank you for the comment.
Point 7. In the 5. Discussion and conclusions, more references to world literature are proposed. Please complete this and the discussion will be more interesting and valuable from a scientific point of view.
Response 7: Thank you for this comment. As stated in the text, to date, the discussed topic has not been addressed by published studies or practical guides and the approach developed by the Authors is innovative and it elaborates upon their preliminary research. It was the only study that relied on GIS tools, in particular Voronoi diagrams, to identify active ageing places. The authors tried to refer to the most relevant literature on senior citizen friendly spaces and GIS applications and Voronoi diagrams in Section 2. In section 5 Discussion and Conclusions references considering the consistency of our research with the current state of international studies in this field have been added. The results obtained in the article extend the existing body of knowledge on possibilities of application of Voronoi diagrams in socio-economic and geographical studies – references to some of those studies were added to the text.

Reviewer 2 Report
The research work is very interesting and could be useful for local authorities to understand the accessibility for some facilities.
Two main issue:
1. For the most part, it was well written. There are needs to proof reading and possibly re-organise some parts of the paper. For example, some issues:
Abstract has small titles, why?
Figure 2 could add some arrow signs to indicating the flow?
Figure 3. there are 2 copies
Table4 very long, poor formatting
Table 5. Active ageing groups and places in the city. Is this table from [97]? If yes, do you need have the whole table here?
2. For the Voronoi diagrams, it would be nice to have more details.
It is not clear how to get the umber of objects per km2 and how the result were aggregated.
A Voronoi polygon is not the best choice of representing the continuous feature (e.g. the accessibility has sudden change in the polygon boundary). Have you considered to smooth these?
And why choose 5-grade scale?
Author Response
Responses to the review of the article entitled:
Voronoi diagrams for senior-friendly cities
submitted to the IJERPH
Upon the Editorial Board’s request, we have made every effort to account for the Reviewers' comments in the revision process. We hope that the article meets the expectations of the Reviewers and the Editorial Board and that it is acceptable for publication in the IJERPH in its present form.
The Authors would like to thank the Reviewers for the time and effort invested in the revision process. The Reviewers’ insightful comments have significantly improved the manuscript’s potential to advance scientific knowledge. We hope that the revised manuscript is suitable for publication.
The responses to the Reviewers’ comments are marked in red. A version of the manuscript in the track changes mode is attached to the resubmission.
Best regards,
The Authors
Response to Reviewer 2 Comments
Point 1. Abstract has small titles, why?
Response 1: Authors have decided to divide the abstract into such parts to emphasise the parts an abstract should consist of.
Point 2. Figure 2 could add some arrow signs to indicating the flow?
Response 2: As the Reviewer rightly suggested, arrows have been added.
Point 3. Figure 3. there are 2 copies.
Response 3: Apologies for the obvious mistake. The unnecessary figure has been removed.
Point 4. Table 4 very long, poor formatting.
Response 4: On the basis of the Reviewer’s remark, the table has been edited (and divided into parts) so it looked more readible to the potential reader.
Point 5. Table 5. Active ageing groups and places in the city. Is this table from [97]? If yes, do you need have the whole table here?
Response 5: No, this table is only based on the information from this reference. Basing on the Reviewer’s suggestion, to improve the quality of editing, it has been changed (divided into two parts).
Point 6. It is not clear how to get the number of objects per km2 and how the result were aggregated.
Response 6: Thank you for this comment. Each object was assigned to one (and one only) Voronoi cell – as in the process of creating Voronoi diagrams. Then, this number (one) was divided by the area of a cell (in km2), giving number of objects per 1 km2 (and the density of objects in the area), i.e. 1 object in a 2-square-kilometer cell means that there are 0.5 objects per 1 km2; or if 1 object in a 0.5-square-kilometer cell means that there are 2 objects per 1 km2. Resulting cells were aggregated into five groups (on a 5 grade scale) basing on their density. These information have been added to the text to clarify this issue.
Point 7. A Voronoi polygon is not the best choice of representing the continuous feature (e.g. the accessibility has sudden change in the polygon boundary). Have you considered to smooth these?
Response 7: The Authors have decided to choose this method as it can be used for preliminary analyses of spatial phenomena (such as proposed active ageing places). What is more, these visualisations are closer to the human perception of space, basing on a more “raw” data. These information are included in section 2 of the article (description of Voronoi diagrams). On the basis of the previous research, Voronoi diagrams provide a much faster product while computing the data, because they result in vector and not raster visualisations. These information have been added to the article, to answer the Reviewer’s accurate comment.
Point 8. And why choose 5-grade scale?
Response 8: The 5-grade scale was chosen on the basis of the obtained results, so that the differences between cell aggregates were visible on maps.

Round 2
Reviewer 2 Report
Thanks for making the effort to revise the paper. I would like to suggest that the following issues should be clarified and discussed with more details:
1. Missing the information about the selected public amenities. You could show those points in the maps of Figure 5.
2. Spatial accessibility could not properly measured using Voronoi diagrams. Voronoi diagrams show the proximity which is not same as accessibility. The 5-grade colour of each cell could not reflect the accessibility. This needs to be made clear.
Author Response
Responses to the review of the article entitled:
Voronoi diagrams for senior-friendly cities
submitted to the IJERPH
Upon the Editorial Board’s request, we have made every effort to account for the Reviewers' comments in the revision process. We hope that the article meets the expectations of the Reviewers and the Editorial Board and that it is acceptable for publication in the IJERPH in its present form.
The Authors would like to thank the Reviewers for the time and effort invested in the revision process. The Reviewers’ insightful comments have significantly improved the manuscript’s potential to advance scientific knowledge. We hope that the revised manuscript is suitable for publication.
The responses to the Reviewers’ comments are marked in red. A version of the manuscript in the track changes mode is attached to the resubmission.
Best regards,
The Authors
Response to Reviewer 2 Comments
Point 1. Missing the information about the selected public amenities. You could show those points in the maps of Figure 5.
Answer 1. The points have been added.
Point 2. Spatial accessibility could not properly measured using Voronoi diagrams. Voronoi diagrams show the proximity which is not same as accessibility. The 5-grade colour of each cell could not reflect the accessibility. This needs to be made clear.
Answer 2. Each colour in the 5-grade scale indicates, how many objects are accessible in the chosen region (Voronoi cell). As stated in section 3, “Voronoi diagrams support the determination of a region’s proximity to a selected set of points, facilitate the identification of the closest public amenities and analyses of their accessibility within the city area.” Each point laying within the Voronoi cell is accessible from this region. That’s why the term “spatial accessibility” was used – just like in:
- Klein, R., Langetepe, E., & Nilforoushan, Z. (2009). Abstract Voronoi diagrams revisited. Computational Geometry, 42(9), 885-902;
- Aggour, H., & Mabrouk, A. (2019, July). Distributed Spatio-Temporal Voronoi Diagrams: State of Art and Application to the Measurement of Spatial Accessibility in Urban Spaces. In International Conference on Artificial Intelligence and Symbolic Computation (pp. 100-117). Springer, Cham;
- Mabrouk, A., Boulmakoul, A., & Bielli, M. (2009). Fuzzy spatial network voronoi diagram: a spatial decision support for transportation planning. International Journal of Services Sciences, 2(3-4), 265-280;
- Cui, H., Wu, L., Hu, S., & Lu, R. (2021). Measuring the service capacity of public facilities based on a dynamic voronoi diagram. Remote Sensing, 13(5), 1027;
- Ai, T., Yu, W., & He, Y. (2015). Generation of constrained network Voronoi diagram using linear tessellation and expansion method. Computers, Environment and Urban Systems, 51, 83-96.
